# Circulating Inflammatory Biomarkers in Early Prediction of Stroke-Associated Infections

**DOI:** 10.3390/ijms232213747

**Published:** 2022-11-09

**Authors:** Isabel M. C. Hasse, Gerrit M. Grosse, Ramona Schuppner, Till Van Gemmeren, Maria M. Gabriel, Karin Weissenborn, Ralf Lichtinghagen, Hans Worthmann

**Affiliations:** 1Department of Neurology, Hannover Medical School, 30625 Hannover, Germany; 2Institute of Clinical Chemistry, Hannover Medical School, 30625 Hannover, Germany

**Keywords:** CRP, IL-6, LBP, IL-10, infection, inflammation, pneumonia, stroke, outcome

## Abstract

(1) Background: Patients with acute ischaemic stroke (AIS) are at high risk for stroke-associated infections (SAIs). We hypothesised that increased concentrations of systemic inflammation markers predict SAIs and unfavourable outcomes; (2) Methods: In 223 patients with AIS, blood samples were taken at ≤24 h, 3 d and 7d after a stroke, to determine IL-6, IL-10, CRP and LBP. The outcome was assessed using the modified Rankin Scale at 90 d. Patients were thoroughly examined regarding the development of SAIs; (3) Results: 47 patients developed SAIs, including 15 lower respiratory tract infections (LRTIs). IL-6 and LBP at 24 h differed, between patients with and without SAIs (IL-6: *p* < 0.001; LBP: *p* = 0.042). However, these associations could not be confirmed after adjustment for age, white blood cell count, reduced consciousness and NIHSS. When considering the subgroup of LRTIs, in patients who presented early (≤12 h after stroke, *n* = 139), IL-6 was independently associated with LRTIs (OR: 1.073, 95% CI: 1.002–1.148). The ROC-analysis for prediction of LRTIs showed an AUC of 0.918 for the combination of IL-6 and clinical factors; (4) Conclusions: Blood biomarkers were not predictive for total SAIs. At early stages, IL-6 was independently associated with outcome-relevant LRTIs. Further studies need to clarify the use of biochemical markers to identify patients prone to SAIs.

## 1. Introduction

Stroke is the second leading cause of death and the third leading cause of disability-adjusted life years [1]. In patients who had a stroke, infections are a predominant risk factor for the outcome and mortality. Stroke-associated infections (SAIs) are defined as infections within the first 7 days after the event and are a common complication of stroke. The most common infections, following a stroke are pneumonia and a urinary tract infection (UTI), with pneumonia in particular being associated with a worse clinical outcome [2,3].

In ischaemic stroke, an inflammatory cascade involving innate and cellular inflammations, as well as oxidative stress, promotes secondary tissue destruction. The inflammatory response interacts with the immune system via the sympatho-adrenergic axis and eventually results in stroke-associated immunodepression, which is suggested to decisively contribute to systemic infections from a local bacterial colonisation [4]. Of note, there is a direct association between the severity and size of the stroke and the degree of inflammation and immunodepression [5].

Randomised controlled trials investigated a general use of antibiotic therapy for the prevention of SAIs and the impact on the clinical outcome and mortality, mostly with neutral results [6,7,8]. Only the Mannheim infection in stroke study (MISS) showed fewer infections and a better outcome for stroke patients treated with meslozillin and sulbactam [9]. Therefore, it is considered that biomarker-based identification of patients, who are prone to SAIs, could be useful for the stratification of patients who might benefit from antibiotic prophylaxis. Smaller studies demonstrated that certain biomarkers could potentially indicate the inflammatory and anti-inflammatory response [10,11].

In a pilot study, we investigated the association of the biomarkers interleukin-6 (IL-6), C-reactive protein (CRP), interleukin-10 (IL-10) and lipopolysaccharide-binding protein (LBP) with SAIs [12]. The pro-inflammatory and acute-phase proteins IL-6 and CRP are important players in the inflammatory cascade and have been shown to be elevated after a stroke. IL-10 has anti-inflammatory properties and is regarded as a mediator of stroke-induced immunodepression. LBP is a biomarker for bacterial lipopolysaccharides and their inflammatory response [13,14]. A number of clinical studies, as well as our pilot study, have suggested an association between these markers and post-stroke infections [10,12,13,15,16].

In this study, we hypothesised that the temporal pattern of CRP, IL-6, IL-10 and LBP would differ between patients with and without SAIs. We aimed to investigate whether these biomarkers might be useful in predicting SAIs and the outcome, in addition to the clinical variables.

## 2. Results

### 2.1. Patients’ Characteristics and Rate of Infection

The study population comprises 223 patients; 37.7% were female, the median age was 74 years (interquartile range 64–81 years) and the median National Institutes of Health Stroke Scale (NIHSS) at admission was 5 (interquartile range 2–10). The demographic and clinical characteristics are reported in Table 1.

Forty-seven patients developed SAIs, including 15 lower respiratory tract infections (LRTIs) and 20 UTIs, two flu-like infections, one erysipelas, one locally infected leg ulcer, one genital candida infection, one gastroenteritis caused by the norovirus, one cholecystitis and five systemic infections without a clear origin, defined by the systemically elevated inflammation mediators, such as CRP > 30 mg/L and/or white blood cell count (WBC) > 11.000/µL in combination with the clinical symptoms, such as fever.

Patients were grouped, according to their status of SAIs (Table 1). The clinical factors that differed significantly between patients with and without SAIs were age (*p* = 0.035), stroke severity (NIHSS upon admission) (*p* < 0.001), dysphagia (*p* < 0.001) and status of consciousness upon admission (*p* < 0.001).

### 2.2. Biochemical Markers of the Inflammation and Status of SAIs

Concentrations of inflammatory markers differed significantly between patients with and without SAIs. Upon admission, IL-6 and LBP were shown to be significantly higher in patients with SAIs (IL-6 24 h: *p* < 0.001; LBP 24 h: *p* = 0.05) while IL-10 (*p* = 0.301) and CRP (*p* = 0.148) did not differ significantly (Table 1).

When analysing the time courses of biomarkers (*n* = 192; excluding patients with missing values, caused by death before seven days (d) or blood sampling errors), levels of IL-6 and LBP differed between the patients with and without SAIs at 24 h and 7d (IL-6 24 h: *p* < 0.001, 3d: *p* < 0.001, 7d: *p* < 0.001; LBP 24 h: *p* = 0.042, 3d: *p* < 0.001, 7d: *p* < 0.001). The CRP differed at 3d and 7d (24 h: *p* = 0.145, 3d: *p* < 0.001, 7d: *p* < 0.001). IL-10 did not differ significantly with respect to the status of SAIs (24 h: *p* = 0.584, 3d: *p* = 0.478, 7d: *p* = 0.220) (Figure 1).

To investigate the association between the biomarkers IL-6 and LBP upon admission and the status of SAIs, we performed a binary logistic regression analysis, defining the clinical parameters, such as age, dysphagia, reduced consciousness at 24 h and stroke severity, as covariables (*n* = 223, no missing values). NIHSS, reduced consciousness and dysphagia were found to be independently associated with SAIs, while IL-6 was not (age: odds ratio (OR): 1.024, 95% confidence interval (CI): 0.991–1.058, *p* = 0.152; reduced consciousness 24 h: OR: 3.148, 95% CI: 1.072–9.246, *p* = 0.037; dysphagia: OR: 2.571, 95% CI: 1.009–6.551, *p* = 0.048; NIHSS: OR: 1.095, 95% CI: 1.012–1.185, *p* = 0.024; IL-6 24 h: OR: 1.010, 95% CI: 0.992–1.029, *p* = 0.284). Similar to IL-6, LBP was also not independently associated with SAIs (OR: 1.084, 95% CI: 0.977–1.203), while the clinical parameters showed an independent association with the occurrence of SAIs (NIHSS, reduced consciousness, dysphagia) (age: OR: 1.020, 95% CI: 0.986–1.054, *p* = 0.251; reduced consciousness 24 h: OR: 3.258, 95% CI: 0.092–9.719, *p* = 0.034; dysphagia: OR: 2.550, 95% CI: 1.008–6.451, *p* = 0.048; NIHSS: OR: 1.097, 95% CI: 1.013–1.187, *p* = 0.022; LBP 24 h: OR: 1.084, 95% CI: 0.977–1.203, *p* = 0.182).

To compare the predictive ability of the combination of the biomarkers and the clinical parameters with the prediction of the clinical parameters alone, the receiver operating characteristic (ROC) statistics deriving from the regression analyses were calculated. The clinical parameters alone already revealed a good discrimination between SAI vs. no SAI (area under the curve (AUC) = 0.810, 95% CI: 0.7372–0.8824), while the combination with IL-6 (AUC = 0.814, 95% CI: 0.7420–0.8851) or LBP (AUC = 0.824, 95% CI: 0.7571–0.8902), respectively, only slightly improved the predictive ability of the corresponding models (ΔAUC + IL-6 = 0.004 (95% CI: −0.0050–0.0125) and ΔAUC + LBP = 0.014 (95% CI: −0.0081–0.0357). (Figure 2)

### 2.3. Association of the Biochemical Markers of Inflammation with the Outcome

To test the hypothesis that the elevated plasma levels of biomarkers of infection are associated with the outcome at 90d, patients were grouped, according to the modified Rankin scale (mRS) (favourable outcome: mRS 0–3; unfavourable outcome: mRS 4–6) (Table 2). In a univariate analysis, IL-6 ≤ 24 h (*p* < 0.001), CRP ≤ 24 h (*p* = 0.034) and LBP ≤ 24 h (*p* = 0.039) differed significantly between patients with a favourable and unfavourable outcome. The logistic regression analysis, adjusted for age, the NIHSS, reduced consciousness, dysphagia and the status of SAIs, failed to show an independent association for the biomarkers ≤24 h with the outcome. The NIHSS, upon admission, was found to be independently associated with the outcome, along with the status of SAIs, age and dysphagia. (For IL-6 (age: OR: 1.133, 95% CI: 1.070–1.200, *p* < 0.001; SAIs within first week: OR: 5.447, 95% CI: 1.801–16.479, *p* = 0.003; reduced consciousness at 24 h: OR: 3.015, 95% CI: 0.732–12.418, *p* = 0.126; NIHSS 24 h: OR: 1.139, 95% CI: 1.034–1.255, *p* = 0.008; dysphagia: OR: 7.957, 95% CI: 2.571–24.621, *p* < 0.001; IL-6: OR: 0.988, 95% CI: 0.951–1.026, *p* = 0.524). For LBP (age: OR: 1.128, 95% CI: 1.065–1.191, *p* < 0.001; SAIs within the first week: OR: 4.718, 95% CI: 1.601–13.901, *p* = 0.005; reduced consciousness at 24 h: OR: 3.231, 95% CI: 0.768–13.603, *p* = 0.110; NIHSS 24 h: OR: 1.139, 95% CI: 1.035–1.253, *p* = 0.008; dysphagia: OR: 7.152, 95% CI: 2.371–21.577, *p* < 0.001; LBP: OR: 1.090, 95% CI: 0.942–1.261, *p* = 0.248). For CRP (age: OR: 1.125, 95% CI: 1.065–1.188, *p* < 0.001; SAIs within the first week: OR: 4.752, 95% CI: 1.625–13.897, *p* = 0.004; reduced consciousness at 24 h: OR: 3.113, 95% CI: 0.759–12.773, *p* = 0.115; NIHSS 24 h: OR: 1.138, 95% CI: 1.034–1.253, *p* = 0.008; dysphagia OR: 6.947, 95% CI: 2.303–20.956, *p* = 0.001; CRP: OR: 1.076, 95% CI: 0.955–1.213, *p* = 0.230)

### 2.4. Association of the Type of Infection with the Outcome

We analysed whether the type of infection was associated with the patient outcome (mRS 90d). In a binary logistic regression analysis (covariables: age, NIHSS at 24 h, reduced consciousness at 24 h, dysphagia), we found that LRTIs were associated with a poor outcome, while UTIs were not. (LRTI within first week: OR: 17.540, 95% CI: 2.184–140.887, *p* = 0.007; UTI within first week: OR: 1.092, 95% CI: 0.221–5.405, *p* = 0.914; Age: OR: 1.154, 95% CI: 1.083–1.230, *p* < 0.001; reduced consciousness at 24 h: OR: 3.522, 95% CI: 0.909–13.648, *p* = 0.068; NIHSS 24 h: OR: 1.158, 95% CI: 1.052–1.275, *p* = 0.003; dysphagia: OR: 6.572, 95% CI: 2.208–19.562, *p* = 0.001).

### 2.5. Early Levels of Inflammation Markers ≤12 h in Association to LRTIs

Since inflammatory markers increase early after a stroke, as shown in our previous pilot study, we intended to characterise the patients who had biochemical marker levels ≤ 12 h (*n* = 139) and stratify the association with LRTIs, as these infections had been shown to be independently associated with the outcome. A univariate analysis revealed that IL-6 (*p* = 0.030), and the clinical parameters NIHSS at 24 h (*p* < 0.001), dysphagia (*p* < 0.001) and reduced consciousness at 24 h (*p* = 0.002) differed significantly between patients with and without LRTIs, while LBP did not (*p* = 0.265). In the multivariable analysis including dysphagia, stroke severity and reduced consciousness at 24 h as covariables, levels of IL-6 were significantly associated with the development of LRTIs. (IL-6: OR: 1.073, 95% CI: 1.002–1.148, *p* = 0.044; reduced consciousness at 24 h: 1.729, OR: 5.635, 95% CI: 0.811–39.167, *p* = 0.080; dysphagia: N.A. since all patients with LRTIs also had dysphagia; NIHSS 24 h: OR: 1.000, 95% CI: 0.869–1.151, *p* = 0.999). Following the identification of IL6 as a significant predictor of LRTIs, a ROC analysis was performed and compared with the predictive value of the clinical parameters for LRTIs (Figure 3). The AUC for the prediction of LRTIs was 0.685 (95% CI: 0.4965–0.8742) for IL-6 and 0.887 (95% CI: 0.8171–0.9558) for the clinical parameters (NIHSS, dysphagia, reduced consciousness, age). The ROC analysis for the combination of clinical parameters and IL-6 yielded an AUC of 0.918 (95% CI: 0.8559–0.9800). Thus, the combination of the clinical variables with the inflammatory biomarker IL-6 was shown to be slightly, yet not significantly, better than the clinical parameters alone in predicting LRTIs (Δ AUC: 0.0312 (95% CI: −0.0227–0.0857)).

## 3. Discussion

The main findings of this study are that (i) the temporal patterns of serum levels of the biomarkers IL-6, LBP and CRP differ significantly between patients with and without SAIs, (ii) the additional predictive value of the circulating biomarkers, in addition to the clinical SAI-predictors, is limited and (iii) in patients following an acute stroke, the serum levels of IL-6 within 12 h of the stroke onset are predictive of LRTIs, which is associated with an unfavourable long-term outcome.

Acute stroke triggers an immunological cascade that leads to an inflammatory response on the one hand and to a systemic immunodepression, on the other, which favours the occurrence of infections and causes a local bacterial colonisation or risk factors, such as bacteriuria or aspiration, leading to an increased risk for systemic infections [4]. It is noteworthy that such an inflammation is also reflected in the peripheral blood by certain biomarkers, which could thus serve as predictors of SAIs [10,11]. An increase in the proinflammatory markers IL-6 and CRP has previously been observed in the serum of patients who suffered an acute stroke [11,16,17]. However, increases in these biomarkers have repeatedly been shown to be associated with a systemic infection as well. Therefore, it is not possible to sharply separate the causes of the increase in the biomarkers, in terms of their association with the response to the ischaemic lesion or with the infection that develops later. For this reason, we paid particular attention to the absence of clinical signs of infection at 24 h, so that the increase in the biomarkers was mostly attributable to the inflammatory response to the stroke at that time point.

In our study, the time course of the pro-inflammatory markers differed between patients with and without SAIs. There was a difference in IL-6 between patients with and without SAIs over the entire time course, from 24 h to 7d. LBP also showed a significant elevation at all time points. CRP showed a delayed increase, and the concentrations were significantly elevated from three to seven days. Strikingly, no difference in serum levels of IL-10 was observed between patients with and without SAIs at any time point in our study. One explanation for this may be that IL-10 levels peak early after a stroke, so the increase may have been missed. For example, our pilot study [12] showed a significant difference in the time course of IL-10 between patients with and without an infection at an earlier time point than 24 h. Another study described that the increased production of IL-6 begins as early as 4–6 h after a stroke and peaks after about 12 h, before dropping off again [11]. Therefore, it can be speculated that in the current study, a time point for the biomarker sampling was chosen that is too late to detect all potential biomarker differences, in regard to incident SAIs. However, stroke patients regularly present later than in the hyperacute setting and the restriction to patients with very early blood sampling would thus come along with limitations, regarding generalisability.

In the current study, we could not show an independent association of the early biomarker levels at <24 h with the occurrence of SAIs and no independent association with the patient outcome after the adjustment for the clinical predictors. Regarding CRP, the evidence is conflicting: Fluri et al. [17] described CRP as a suitable predictor for the occurrence of SAIs. In contrast, Wartenberg et al. [10] stated that the CRP increase occurs at a later time point and that the biomarker should be considered as a sign of infection rather than a predictive marker. This reasoning is also supported by the results of our study.

Overall, the study showed that the clinical parameters, such as the NIHSS, dysphagia, age and a reduced consciousness alone had a good predictive value for the occurrence of SAIs and that the combination of the biomarkers with clinical parameters only slightly improves this predictive ability.

In accordance with these findings, Hotter et al. showed no significant improvement in the predictive power when biomarkers and clinical variables were combined, whereas other biomarkers were considered (e.g., procalcitonin, copeptin or C-terminal pro-endothelin-1) [18]. In contrast, Salat et al. described a significant improvement in the predictive power when clinical factors were combined with the biomarkers IL-13 and interferon (IFN)-γ [19].

The selection of clinical parameters in our study that act as predictive parameters for the occurrence of infections, is also reflected in the literature. In a systematic review by Meisel et al., the clinical risk scores for the occurrence of stroke-associated pneumonia were compared. The authors summarised that age, the NIHSS, dysphagia and reduced consciousness were the most important clinical parameters [19].

In accordance with the current literature, our study confirms that the occurrence of SAIs is associated with an unfavourable outcome [20,21]. Interestingly, it also shows that not every type of infection has the same impact on the outcome. In contrast to LRTIs, UTIs were not associated with the neurological outcome.

Westendorp et al. [3] demonstrated a significant impact of both LRTIs and UTIs on a stroke outcome, but also stated that LRTIs in particular, were associated with a higher mortality. Bustamante et al. showed that respiratory infections were among the stroke complications that had a high impact on in-hospital mortality and suggested that inflammatory and immunological biomarkers may be useful for selecting patients at risk for pneumonia [22]. This could possibly be due to the fact that respiratory worsening can become immediately life-threatening, especially in bedridden, frail patients who cannot expectorate vigorously and in whom sepsis can also develop quickly. Moreover, the systemic inflammatory response in patients with LRTIs is likely stronger than in patients with UTIs which may have additional implications on the neuroinflammation and thus the functional outcome. Based on the result that LRTIs particularly predispose for an unfavourable outcome, we performed a subgroup analysis.

In this subgroup, IL-6 deriving from the early serum samples was shown to be associated with the development of LRTIs independently of the clinical variables and thus could be used as a predictive marker. However, again, the additive predictive value of this biomarker was limited, compared to the already excellent value of the clinical predictors alone.

In accordance, a study by Faura et al. showed an association of IL-6 with the occurrence of respiratory tract infections and that the combination with biomarkers (in this case IL-6, von Willebrand factor (vWF) and D-dimer) had superior predictive power than the clinical values alone [23]. The better result of the combination of IL-6 and the clinical variables may show the additional benefit of IL-6 by capturing patients with an increased severity, such as the association with stroke size, which is not always reflected by the clinical scores e.g., NIHSS [11].

In recent years, a number of studies have been conducted aiming to prevent SAIs by anti-infective antibiotic therapy, in order to improve the clinical outcomes. However, while a general anti-infective therapy partially reduced the incidence of SAIs, the outcomes remained largely unaffected [6,8]. As in other studies, SAIs were confirmed to be associated with a worse outcome and a higher mortality, and LRTIs in particular, were associated with a poor outcome [21].

In the Mannheim infection stroke study (MISS), it was shown that antibiotic prophylaxis can nevertheless have a positive effect on the outcome [9]. Significantly fewer infections occurred under antibiotic therapy and also the outcome after 90 days was improved. The results of the MISS study therefore indicate that further risk stratification is needed in order to identify patients who may benefit from an antibiotic preventive treatment.

To clarify the effectiveness of the preventive antibiotics for post-stroke infections, a meta-analysis was conducted that showed a significant reduction in SAIs but no reduction in mortality or improvement in outcomes. Nevertheless, the results of this analysis suggested that a prior identification of patients who would benefit from antibiotic administration, may be useful [8].

This identification could take place, for example, as mentioned above, by identifying patients who have a higher probability of contracting an outcome-relevant SAI and would therefore benefit from antibiotic prophylaxis. It is conceivable that future studies on the prevention of LRTIs may need to investigate the interventions and medications other than antibiotics. This is because antibiotics are helpful for the acute therapy of pneumonia but may not be the best means of prevention. It is possible that the prophylactic administration of antibiotics may lead to the development of resistant strains of bacteria via the partial depletion of the gut microbiome, which in turn are more difficult to treat, as shown in an animal model [24]. In addition to antibiotics, there are also further approaches to reduce the risk of SAIs, for example utilising metoclopramide [25] or beta-receptor inhibition [26].

In summary, the combination of biomarkers and clinical factors did not substantially increase the predictive power of SAIs, compared to the clinical parameters alone. However, IL-6 could act as a predictive marker for the occurrence of LRTIs and could possibly be used in combination with the clinical parameters of other biomarkers to identify patients who might benefit from antibiotic therapy. Further studies are merited to elaborate on the potential value of circulating biomarkers in prediction of SAIs and their subtypes.

Emphasis should be placed on LRTIs that are relevant to the clinical outcome. In addition, a larger number of cases should be studied to perform a subgroup analysis, particularly allowing for stroke etiology analysis. Thus, in the current study, it remains unclear what significance should be attributed to the absence of SAIs in patients with a small vessel occlusion. It could either be related to the lower clinical severity in these patients or have specific immunologic findings (for a review of this etiology, see [27]).

Our study contains major limitations, which should be discussed. Considering the number of covariates in the multivariable models, the sample size is still low. Of note, the incidence of 47 out of 223 patients with SAIs (approx. 21%) in the current study corresponds to the literature. In addition, the study was conducted using a single center approach. Another limitation might be the selection of the biomarkers in this study, because other biomarkers might be more sensitive for the detection of immunodepression, e.g., procalcitonin or the soluble triggering receptor expressed on myeloid cells-1 (s-TREM-1), which have been widely used as markers for infection or even sepsis [28,29]. However, the selection of biomarkers in the current study was mandatory, as it was bound to follow the selection in the pilot study [12]. It should be noted that we did not recruit patients consecutively because of the inclusion and exclusion criteria and, most importantly, because of an early timing of blood collection after informed consent, which was competitive with sometimes limited availability of stay personnel, for example, outside working days. However, the baseline characteristics of our study collective are clearly representative for the entirety of stroke patients in our hospital.

## 4. Materials and Methods

### 4.1. Study Population

From August 2014 to November 2018 a total of 461 patients were prospectively screened for the study inclusion at the Neurological Clinic of Hannover Medical School. Following the consideration of the inclusion and exclusion criteria, 223 patients with an acute ischaemic stroke with symptom onset within the last 24 h could be considered. The exclusion criteria are shown in Figure 4. Diagnosis of an acute ischaemic stroke was defined by the radiological evidence of infarction (cranial magnetic resonance imaging (cMRI) or cranial computed tomography scan (cCT)) and reliable clinical symptoms.

Within the first seven days after symptom onset, patients were prospectively investigated for infection by experienced physicians, using clinical examinations and appropriate diagnostics, if mandatory (e.g., chest X-ray, urine examination). Diagnosis of an infection was based on the recommendations from the Pneumonia in Stroke Consensus Group and the Centers for Disease Control and Prevention (CDC) and the National Healthcare Safety network (NHSN) surveillance definition of healthcare-associated infections [30,31].

Upon admission, clinical and demographical data of the patients were collected, including age, sex, localisation and aetiology of stroke, using the Trial of Org 10,172 in Acute Stroke Treatment (TOAST) classification, cardiovascular risk factors, such as arterial hypertension, diabetes mellitus, nicotine consumption, hyperlipidemia, glomerular filtration rate and the Essen stroke risk score (ESRS). The stroke severity was determined using the NIHSS. The patients’ functional status prior to the index event of stroke was assessed using the pre-morbid modified Rankin Score (pre-mRS). The outcome at 90 days was determined by a clinical follow-up visit applying the NIHSS, the mRS and the Barthel Index (BI).

### 4.2. Blood Samples and Marker Determination

Serum, heparin plasma and EDTA plasma were collected at the following time points: ≤24 h of symptom onset and on day 3, 7 and 90 after the index event of the stroke. Following the collection, the blood was centrifuged immediately, and stored at −80 °C until further processing for measurements.

Commercially available CE (Communautés Européennes)-certified reagents for the clinical laboratory diagnostics were used to measure the biochemical parameters, following the manufacturers’ instructions. IL-6, LBP and IL-10 were measured using the Immulite 1000 (Siemens Healthcare GmbH, Eschborn, Germany) and CRP using the Atellica NEPH 630 System (Siemens Healthcare GmbH, Eschborn, Germany). IL-6 and CRP were measured in serum samples, LBP in EDTA-plasma and IL-10 in heparin-plasma.

### 4.3. Statistics

The analysis of the data was performed using SPSS software package version 26 (IBM-Deutschland, Munich, Germany), SAS Enterprise Guide 7.1 (SAS Institute Inc., Cary, North Carolina, USA) and SigmaPlot 12.0 (Systat Software GmbH, Frakfurt am Main, Germany). Patients were divided into two groups, according to status of SAIs. In a univariate analysis, the differences in the baseline characteristics, such as the clinical characteristics, demographics and blood marker levels between stroke patients with and without SAIs were investigated. All variables were tested for the normal distribution using the Kolmogorov Smirnov test and were found not to be normally distributed. Accordingly, the Mann–Whitney U test was used to identify the differences in continuous variables and the Pearson chi-squared test for the categorical variables. The binary logistic regression analyses were performed to assess the association between the concentrations of each respective biomarker at ≤24 h and the occurrence of SAIs with age, the NIHSS upon admission, dysphagia and reduced consciousness upon admission as covariates, using the method of the stepwise backward elimination. Furthermore, the logistic regression was performed to detect an association between biomarker levels and the clinical outcome, as determined by mRS at 90d. An unfavourable outcome was defined as mRS of 4–6. In a subgroup analysis of patients with blood samples taken ≤12 h after symptom onset, the association between IL-6 and SAIs was tested in a logistic regression analysis. OR for the prediction of SAIs, they were calculated with 95%-CI per unit-increase of biomarkers. The ROC-curves and the corresponding AUC deriving from the regression models were computed. Values of *p* < 0.05 were considered as statistically significant. A Sigma Plot Systat Software GmbH, Frakfurt am Main, Germany) was used to visualise the time course of the biomarkers in patients with and without SAIs. The ROCs were created using SAS Enterprise Guide 7.1 (SAS Institute Inc., Cary, North Carolina, USA).

## 5. Conclusions

In conclusion, the selected biomarkers IL-6, IL-10, LBP and CRP were shown to have no substantial additive predictive value for the entirety of SAIs, compared to clinical variables alone. However, early serum concentrations of IL-6 have been shown to have a significant association with the occurrence of LRTIs, independent of other clinical variables, and these LRTIs in particular have a relevant negative effect on the outcome.

## Figures and Tables

**Figure 1 ijms-23-13747-f001:**
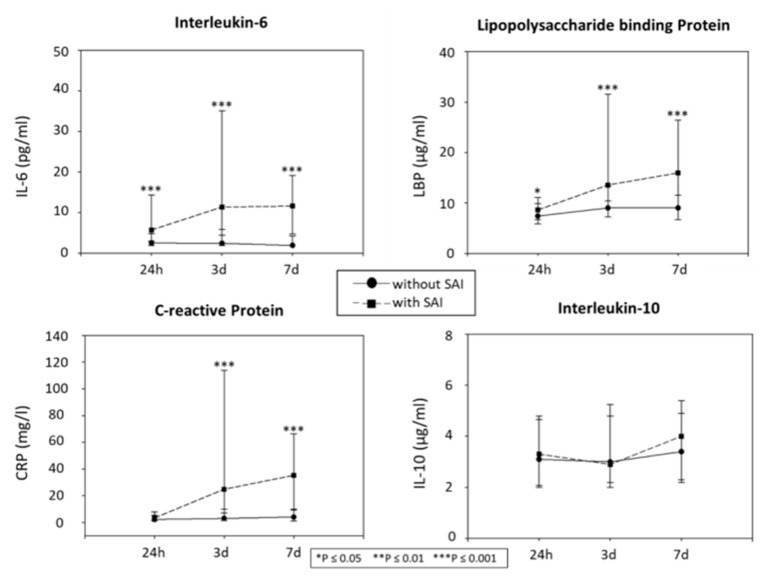
Temporal pattern of the biomarkers in patients with and without SAIs. Biomarker’s temporal pattern in patients with (*n* = 44) and without infection (*n* = 148). * indicates significant differences between the groups (* *p* ≤ 0.05; ** *p* ≤ 0.01; *** *p* ≤ 0.001). IL, Interleukin; LBP, lipopolysaccharide binding protein; CRP, C-reactive protein; h, hours; d, days; SAI, Stroke-associated infection.

**Figure 2 ijms-23-13747-f002:**
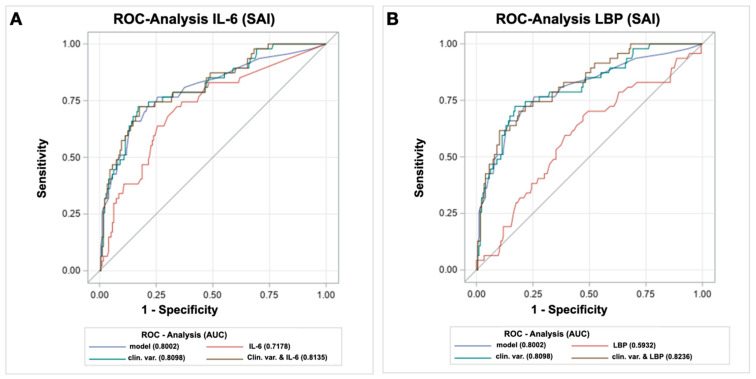
Receiver operating characteristics (ROC)-curves of the baseline biomarkers and the clinical variables in predicting SAIs. (**A**): IL-6; (**B**): LBP. AUC, area under the curve; IL, Interleukin; LBP, lipopolysaccharide binding protein; ROC, Receiver Operating Characteristics; SAI, Stroke-associated infection. Clinical variables comprised the National Institutes of Health Stroke Scale (NIHSS), dysphagia, reduced consciousness and age.

**Figure 3 ijms-23-13747-f003:**
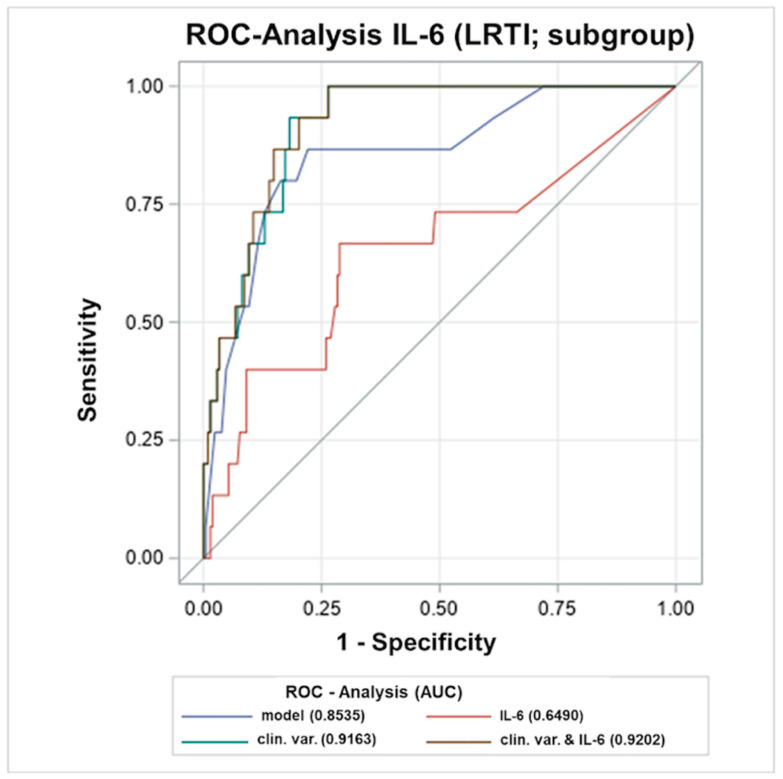
Receiver operating characteristic (ROC)-curves of early (<12 h) IL-6 and clinical variables in predicting LRTIs. AUC, area under the curve; IL, interleukin; LRTI, lower respiratory tract infection; ROC, Receiver Operating Characteristics; SAI, Stroke-associated infection; h, hours. Clinical variables comprised National Institutes of Health Stroke Scale (NIHSS), dysphagia, reduced consciousness and age.

**Figure 4 ijms-23-13747-f004:**
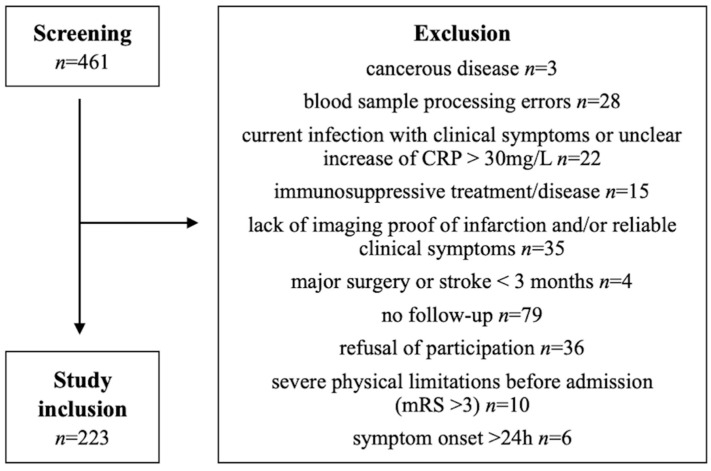
Selection flow chart with the reasons for the study exclusion.

**Table 1 ijms-23-13747-t001:** Distribution of the Clinical Characteristics According to SAI Status.

	All Patients (*n* = 223)	SAI = Yes (*n* = 47)	SAI = No (*n* = 176)	*p*=
Female [%]	84 [37.67]	23 [48.94]	61 [34.66]	0.073
Age in years [IQR]	74 [64.00–81.00]	79 [68.00–84.00]	73 [64.00–80.75]	0.035
BMI [IQR)	26.0 [24.14–28.91]	25.7 [23.80–29.00]	26.1 [24.17–28.90]	0.763
Blood glucose upon admission [IQR]	6.5 [5.7–7.9]	6.8 [6.20–8.50]	6.5 [5.70–7.80]	0.178
NIHSS upon admission [IQR]	5 [2.00–10.00]	13 [6.00–17.00]	4 [2.00–8.00]	<0.001
Reduced consciousness upon admission [%]	26 [11.66]	17 [36.17]	9 [5.11]	<0.001
Dysphagia [%]	76 [34.08]	33 [70.21]	43 [24.43]	<0.001
CHA2DS2-VASc [IQR]	5 [4.00–6.00]	6 [4.00–7.00]	5 [4.00–6.00]	0.089
ESRS [IQR]	4 [3.00–5.00]	4 [3.00–6.00]	4 [3.00–5.00]	0.335
Stroke cause (TOAST)				0.019
TOAST large-artery atherosclerosis [%]	19 [8.52]	2 [4.26]	17 [9.66]	
TOAST cardioembolism [%]	88 [39.46]	25 [53.19]	63 [35.80]	
TOAST small-vessel occlusion [%]	16 [7.17]	0 [0.00]	16 [9.09]	
TOAST other determined etiology [%]	4 [1.79]	2 [4.26]	2 [1.14]	
TOAST undetermined etiology [%]	96 [43.05]	18 [38.30]	78 [44.32]	
Atrial fibrillation [%]	66 [29.60]	19 [40.43]	47 [26.70]	0.067
Coronary heart disease [%]	30 [13.45]	8 [17.02]	22 [12.50]	0.420
Renal dysfunction [%]	50 [22.42]	11 [23.40]	39 [22.16]	0.856
History of stroke [%]	48 [21.52]	13 [27.66]	35 [19.89]	0.249
Family history of stroke [%]	66 [29.60]	10 [21.28]	56 [31.82]	0.160
Obesity (BMI ≥ 30 kg/m^2^) [%]	47 [21.08]	12 [25.53]	35 [19.89]	0.399
Arterial hypertension [%]	172 [77.13]	33 [70.21]	139 [78.98]	0.204
Hyperlipoproteinemia [%]	54 [24.21]	12 [25.53]	42 [23.86]	0.813
Alcohol abuse [%]	25 [11.21]	6 [12.77]	19 [10.80]	0.704
Nicotine abuse [%]	119 [53.36]	27 [57.45]	92 [52.27]	0.528
Diabetes [%]	42 [18.83]	6 [12.77]	36 [20.45]	0.231
IL-6 24 h (pg/mL) [IQR]	3.1 [1.90–6.50]	5.8 [3.10–14.90]	2.6 [1.90–4.68]	<0.001
LBP 24 h (ug/mL) [IQR]	7.8 [6.00–10.40]	8.6 [6.70–11.10]	7.5 [5.93–9.90]	0.05
IL-10 24 h (ug/mL) [IQR]	3.2 [2.00–5.00]	3.5 [2.30–5.90]	3.1 [1.90–4.98]	0.301
CRP 24 h (mg/L) [IQR]	2.9 [1.15–6.63]	3.35 [1.26–8.09]	2.38 [1.15–6.16]	0.148

Note. Statistical analysis was performed using the Pearson chi-squared test for the categorical variables (presented as percentages) and the Mann–Whitney U test for the continuous variables (presented as median with 25th to 75th percentiles). SAIs, Stroke-associated infection; BMI, Body Mass Index; NIHSS, National Institutes of Health Stroke Scale; ESRS, Essen Stroke Risk Score; TOAST, Trial of ORG 10,172 in acute stroke treatment; IL-6, interleukin 6; LBP, lipopolysaccharide binding protein; IL-10, interleukin 10; CRP, C-reactive protein; h, hours. *p* < 0.05 was considered significant, regarding the differences between patients with and without SAIs.

**Table 2 ijms-23-13747-t002:** Distribution of the Patient Characteristics According to the mRS Status at 90d.

	All Patients (*n* = 223)	Favourable Outcome (mRS 0–3) (*n* = 171)	Unfavourable Outcome (mRS 4–6) (*n* = 52)	*p*=
Female [%]	84 [37.67]	57 [33.33]	27 [51.92]	0.013
Age in years [IQR]	74 [64–81]	71 [62–80]	81 [75–87]	<0.001
BMI [IQR)	26.0 [24.1–28.9]	26.2 [24.1–29.1]	25.7 [23.8–27.8]	0.415
Blood glucose on admission [IQR]	6.5 [5.7–7.9]	6.5 [5.6–7.7]	7.5 [6.3–9.3]	0.003
Reduced consciousness on admission [%]	26 [11.66]	6 [3.51]	20 [38.46]	<0.001
Dysphagia [%]	76 [34.08]	34 [19.88]	42 [80.77]	<0.001
NIHSS on admission [IQR]	5 [2–10]	4 [2–7]	13 [5–18]	<0.001
SAI within 7d [%]	47 [21.08]	17 [9.94]	30 [57.69]	<0.001
UTI within 7d [%]	19 [8.52]	9 [5.26]	10 [19.23]	0.004
LRTI within 7d [%]	15 [6.73]	2 [1.17]	13 [25.00]	<0.001
Alcohol abuse [%]	25 [11.21]	20 [11.70]	5 [9.62]	0.448
Atrial fibrillation [%]	66 [29.60]	43 [25.15]	23 [44.23]	0.008
Arterial Hypertension [%]	172 [77.13]	129 [75.44]	43 [82.69]	0.184
IL-6 24 h (pg/mL) [IQR]	3.1 [1.9–6.5]	2.6 [1.9–4.7]	5.6 [2.6–14.9]	<0.001
LBP 24 h (ug/mL) [IQR]	7.8 [6.0–10.4]	7.5 [6.0–9.9]	8.7 [6.3–10.9]	0.039
IL-10 24 h (ug/mL) [IQR]	3.2 [2.0–5.0]	3.2 [1.9–5.0]	3.1 [2.3–5.0]	0.610
CRP 24 h (mg/L) [IQR]	2.39 [1.15–6.63]	2.27 [1.15–4.86]	5.23 [1.15–8.28]	0.034

Note. Statistical analysis of these values was performed using the Pearson chi-squared test for the categorical variables (presented as percentages) and the Mann–Whitney U test for the continuous variables (presented as median with 25th to 75th percentiles). mRS, modified Rankin Scale; BMI, Body Mass Index; NIHSS, National Institutes of Health Stroke Scale; SAIs, Stroke-associated infection; UTI, Urinary tract infection; LRTI, lower respiratory tract infection; d, days; h, hours. *p* < 0.05 was considered significant, regarding the differences between the patients with favourable and unfavourable outcomes at 90d (based on mRS at 90d).

## Data Availability

Not applicable.

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
