# Peer review of "Circulating Inflammatory Biomarkers in Early Prediction of Stroke-Associated Infections"

_ijms, 2022, doi:10.3390/ijms232213747_

Round 1

Reviewer 1 Report

The manuscript is well written and aims to identify early inflammatory markers predictors of stroke-associated infections. The authors should clarify why they chose those IL-6, IL-10, CRP and LBP markers. Why didn't they use procalcitonin and s-TREM1 as markers?

I recommend citing these two articles: DOI: 10.1016 / j.legalmed.2017.07.002 DOI: 10.1177 / 2058738419855226

There are grammatical errors.

Author Response

Point 1:

The manuscript is well written and aims to identify early inflammatory markers predictors of stroke-associated infections. The authors should clarify why they chose those IL-6, IL-10, CRP and LBP markers. Why didn't they use procalcitonin and s-TREM1 as markers?

Response 1:

We thank the reviewer for reading our manuscript and for the valuable feedback.

We have now already referred in the Introduction to other observational clinical studies that investigated the association of said markers with post-stroke infections.

(Introduction section, page 2, paragraph 3)

The restricting selection of markers including the mentioned suggested markers was added in the limitations.

(Discussion section, page 10, last paragraph of discussion.)

Point 2:

I recommend citing these two articles: DOI: 10.1016 / j.legalmed.2017.07.002 DOI: 10.1177 / 2058738419855226

Response 2:

Thank you. The proposed studies have been integrated.

(Discussion section, page 10, last paragraph)

Point 3:

There are grammatical errors.

Response 3:

We have read the paper again thoroughly and corrected grammatical errors as well as mistakes in wording and spelling.

Reviewer 2 Report

The authors studied 223 patients with stroke and analyzed predictive value of early inflammatory responses (IL6, IL10, CRP, LBP) in the group of 21% who got infected afterwards vs the not infected.  The results really could not prove any additive effect of the inflammartory responses in addition to clinicaal routine patient characteristics. Possibly pneuminia could be predicted but this is really a subanalysis in the study.

In conclusion, the study is very ambitious screening of 450 patients in one institution and performing complex multivariate analysis in a seemingly correct way in order to find the clues to this serious disease. A good effort.

I noted misspelling of "hsitory" in one place, and LRTI is not explained in the text body (but in the abstract). Methods are presented after the results, this is according to the journal rules I suppose.

Author Response

Point 1:

The authors studied 223 patients with stroke and analyzed predictive value of early inflammatory responses (IL6, IL10, CRP, LBP) in the group of 21% who got infected afterwards vs the not infected.  The results really could not prove any additive effect of the inflammartory responses in addition to clinicaal routine patient characteristics. Possibly pneuminia could be predicted but this is really a subanalysis in the study.

In conclusion, the study is very ambitious screening of 450 patients in one institution and performing complex multivariate analysis in a seemingly correct way in order to find the clues to this serious disease. A good effort.

Response 1:

We thank the reviewer for reading our manuscript and for the valuable feedback.

Point 2:

I noted misspelling of "hsitory" in one place, and LRTI is not explained in the text body (but in the abstract). Methods are presented after the results, this is according to the journal rules I suppose.

Response 2:

The spelling error in Table 1 has been corrected.

(Table 1, page 3)

The abbreviation LRTI was explained.

(Results section, page 2, paragraph 2)

The Materials and methods section is inserted after the Discussion section, as specified by the Instructions for authors.